# Vitamin D Status in Critically Ill Patients with SIRS and Its Relationship with Circulating Zn and Related Parameters during ICU Stay

**DOI:** 10.3390/nu14173580

**Published:** 2022-08-30

**Authors:** Lourdes Herrera-Quintana, Héctor Vázquez-Lorente, Jorge Molina-López, Yenifer Gamarra-Morales, Javier Ignacio Martín-López, Elena Planells

**Affiliations:** 1Department of Physiology, School of Pharmacy, Institute of Nutrition and Food Technology “José Mataix”, University of Granada, 18071 Granada, Spain; lourdesherrera@ugr.es (L.H.-Q.); jennifer_gamo@hotmail.com (Y.G.-M.); elenamp@ugr.es (E.P.); 2Faculty of Education, Psychology and Sports Sciences, University of Huelva, 21007 Huelva, Spain; 3Intensive Care Unit, PTS-San Cecilio University Hospital, 18007 Granada, Spain; jignacio.martin.sspa@juntadeandalucia.es

**Keywords:** vitamin D, Zinc, critically ill patient, Intensive Care Unit, Systemic Inflammatory Response Syndrome

## Abstract

Critically ill patients are exposed to different stressors which may generate Systemic Inflammatory Response Syndrome (SIRS). This situation hinders the assessment of micronutrients status, such as vitamin D or Zinc (Zn), potentially affecting patients’ treatment and recovery. The aim of the present study was to assess the evolution of circulating 25–Hydroxyvitamin D (25–OH–D) levels after seven days of Intensive Care Unit (ICU) stay and the influence on changes in plasma and erythrocyte Zn levels, as well as other parameters related to phosphorus–calcium metabolism. A prospective analytical study was conducted on 65 critically ill patients (42% women) aged 31–77 years with SIRS. Total 25–OH–D levels were measured in plasma samples by liquid chromatography-tandem mass spectrometry, and Zn content was analyzed by flame atomic absorption spectrometry. Both 25–OH–D and 25–OH–D_3_ levels were directly associated with erythrocyte Zn concentration at follow-up (*p* = 0.046 and *p* = 0.011, respectively). A relationship between erythrocyte and plasma Zn was also found at this follow-up point. No such clear associations were found when considering 25–OH–D_2_. Different disturbances in levels of phosphorus–calcium metabolism parameters were found, suggesting a relationship between the changes of 25–OH–D_3_ levels and parathormone (*p* = 0.019) and phosphorus (*p* = 0.005). The findings of the present study suggest an interaction between vitamin D and Zn, in which the correct status of these micronutrients could be a potentially modifiable factor and a beneficial approach in the recovery of critically ill patients.

## 1. Introduction

Critically ill patients admitted to the Intensive Care Unit (ICU) are exposed to different stressors involved in an acute stress response which may lead to inflammation and metabolic disturbances (e.g., hyperglycemia or mineral imbalances) [1]. An exaggerated host defense response, with the dysregulation of proinflammatory and anti-inflammatory pathway homeostasis, may generate Systemic Inflammatory Response Syndrome (SIRS) [2]. The presence of inflammation in multiple contexts (e.g., surgery, trauma, infection, or many acute or chronic diseases) could influence the assessment of the status of many micronutrients, hence inducing a redistribution from the circulating compartment to other organs [3].

Vitamin D is a micronutrient that acts as a hormone, and is involved in a wide range of biological functions through both an endocrine mechanism (e.g., regulation of Calcium (Ca) absorption) and an autocrine pathway (e.g., facilitation of gene expression) [4]. There are roughly 800 human genes for which there is a vitamin D response element; most of them are related to the expression of proteins necessary for the control of cell proliferation, differentiation, and apoptosis [4,5], and other functions related to the immune system [6] or antioxidative actions [7]. Cholecalciferol or vitamin D_3_ is the natural form of vitamin D in animals and the form synthesized in human skin on exposure to sunlight, this being the most important source of vitamin D [8]. Typical dietary animal sources of vitamin D_3_ are fatty fish, cod liver oil, or egg yolks, and fungal sources, such as mushrooms and yeast exposed to sunlight or UV radiation containing vitamin D_2_ (ergocalciferol) [9]. Vitamin D_3_ is more potent than vitamin D_2_, but is itself not biologically active and needs to be hydroxylated in the liver, forming, D_3_ metabolite, 25–Hydroxyvitamin D_3_ (25–OH–D_3_) (the major circulating form of vitamin D and reliable biomarker of vitamin D status), and later in the kidney, forming 1,25–Dihydroxyvitamin D_3_ (1,25–(OH)_2_–D_3_) [10]. Together, vitamins D_3_ and D_2_ make up total vitamin D.

Zinc (Zn) is an essential mineral for all organisms because it plays an important role in numerous processes, including protein and nucleic acid synthesis, carbohydrate and lipid metabolism, and both innate and adaptive immunity, and it also acts as a cofactor and component for numerous enzymes and transcription factors, among others [11,12]. Flesh foods are the most important dietary common sources of readily bioavailable Zn, being red meat richer in Zn than fowl and fish [13]. Over 85% of total body Zn is found in skeletal muscle and bone, with only a very small amount (0.1% of total) found in plasma [14]. Plasma Zn concentrations decrease during inflammation, largely because of the redistribution of Zn from albumin to the liver [15], which may further influence or even control the intensity of the acute phase response. In this regard, while acute Zn deficiency causes a decrease in innate and adaptive immunity, its chronic deficiency increases inflammation [16].

The relationship between vitamin D and Zn is not well known. Few lines of evidence have described the administration of vitamin D to effectively reduce the severity of Zn deficiency, suggesting an important effect on Zn absorption [17]. Furthermore, adequate circulating 25–OH–D_3_ levels have been associated with improved absorption of essential minerals, including Zn [18]. Zn and vitamin D are important for bone metabolism, their deficiencies being associated with an increased risk of osteoporosis [19]. Zn has been shown to inhibit osteoclastic bone resorption and stimulate bone formation and mineralization [20,21]. Additionally, the structure of the DNA binding domain of the 1,25–OH_2_–D_3_ receptor is modulated by Zn, the activity of vitamin D-dependent genes in cells being influenced by intracellular Zn concentrations [22,23]. Moreover, vitamin D has been demonstrated to generate a massive increase in Zn transport in cultured cells [7], and it is also hypothesized that vitamin D may mitigate oxidative damage by improving intracellular Zn concentrations [24]. Hence, based on the above-mentioned aspects, vitamin D and Zn appear to play key roles in many biological processes, which could be crucial for the recovery of critically ill patients. However, little is known about the combined action of these micronutrients.

The present study aimed to assess the evolution of circulating 25–OH–D levels among critically ill patients with SIRS at admission and after one week of ICU stay, and the influence of these levels on changes in plasma and erythrocyte Zn levels and other parameters related to phosphorus–calcium metabolism. We hypothesized that plasma 25–OH–D levels would be decreased at baseline and follow-up, evidencing changes in Zn circulating levels, among others.

## 2. Materials and Methods

### 2.1. Subjects and Study Design

The present multicenter observational study was carried out on 65 critical ill patients (aged 31–77 years, 27 women) recruited from Hospital Clínico San Cecilio and Virgen de las Nieves Hospital in Granada (Spain), and Santa Ana Hospital in Motril (Spain), during the period from September 2015 to September 2019. The participants were monitored from ICU admission (baseline) until day 7 of stay (follow-up), the mortality rate being 38.5% during the study period. All eligible participants enrolled in the study were critically ill patients and fulfilled the following criteria: (i) aged 18 years or older; (ii) with SIRS [2]; and (iii) who agreed to participate in the study, or in which approval of participation was obtained from the family. Exclusion criteria were: (i) refusal to participate in the study as expressed by the patient or his/her legal representatives; (ii) pregnancy; (iii) the presence of highly contagious disease; (iv) allergies; (v) cancer; and (vi) the ingestion of food before obtaining the analytical sample. The present study was conducted in accordance with the principles of the Declaration of Helsinki (last revised guidelines from 2013) [25], following the International Conference on Harmonization (ICH)/Good Clinical Practice (GCP) standards, and was approved by the Ethics Committee of the University of Granada (Ref. 149/CEIH/2016).

### 2.2. Nutritional Profile

All patients received standard nutritional support via the enteral, parenteral, or combined routes, and were administrated nutritional formulas following the hospital’s protocol. A daily nutritional log was kept for each patient (i.e., type, volume, composition of intake and tolerance) from admission to 7 days in the ICU. Mineral and vitamin support was calculated daily based on the age, weight, and sex of each individual, according to current ESPEN guidelines [26], registered by the nutritionists and represented as the average of the 7-day period of stay in the ICU. The nutritional support protocol in critically ill patients was assessed according to the Clinical Nutrition Units of the Hospitals, based on the American Society for Parenteral and Enteral Nutrition and the European Society of Parenteral and Enteral Nutrition guidelines [26].

### 2.3. Data Collection

The study data, including age, sex, weight, height, Body Mass Index (BMI), calculated as weight (Kg)/height (m^2^), total proteins, albumin, prealbumin, ferritin, transferrin, triglycerides, total cholesterol, C-Reactive Protein (CRP), Gamma-Glutamyl Transferase (GGT), Glutamic Oxaloacetic Transaminase (GOT), Glutamic Pyruvic Transaminase (GPT), Alkaline Phosphatase (ALP), osteocalcin, Parathormone (PTH), and Ca, Magnesium (Mg) and Phosphorus (P) were retrieved from the hospital electronic database system and recorded for each study participant at ICU admission (baseline) and after 7 days (follow-up). Biochemical parameters were determined in the Laboratory Analysis Unit of Virgen de las Nieves Hospital (Granada) (ECLIA, Elecsys 2010 and Modular Analytics E170, Roche Diagnostics, Mannheim, Germany). The Acute Physiology and Chronic Health Evaluation II (APACHE-II) score and Sequential Organ Failure Assessment (SOFA) score were obtained by intensivists in the ICU.

### 2.4. Blood Sampling and Biochemical Parameters

Two blood extractions were performed in the morning under fasting conditions, at ICU admission (baseline) and after one week of ICU stay (follow-up). Plasma was separated through centrifugation (4 °C for 15 min at 3000 rpm) and the erythrocytes were washed 4 times with 3 mL of 0.9% sodium chloride solution, centrifuging for 15 min at 3000 rpm after each wash and removing the supernatant saline solution from the last wash. Samples were stored at −80 °C until analytical determination of the different parameters. The recorded biochemical parameters were determined using routine hospital analytical assays (ECLIA, Elecsys 2010 and Modular Analytics E170, Roche Diagnostics, Mannheim, Germany).

#### 2.4.1. Analytical Determination of Zn

As previously described, Zn content was analyzed by Flame atomic Absorption Spectrometry (FAAS) (Perkin Elmer A. Analyst’300 Norwalk, Connecticut, USA) in the previous wet-mineralized way in the Scientific Instrumental Center (SIC) from the University of Granada [27]. All samples were measured in one run, in the same assay batch and blinded quality control samples were included in the assay batches to assess laboratory error in the measurements. Reference values for plasma Zn concentrations were 10–17 µmol/L (0.65–1.11 mg/L) [14].

#### 2.4.2. Analytical Determination of Vitamin D

The extended protocol has been described previously [28]. Briefly: total 25–OH–D levels were measured in plasma samples by Liquid Chromatography-Tandem Mass Spectrometry (LC-MS/MS). Plasma sample treatment involved protein precipitation, extraction, and derivatization. The Endocrine Society recommends 25–OH–D levels below 20 ng/mL to be termed vitamin D deficiency, concentrations of 21–29 ng/mL to be termed insufficient, and normal levels should be reserved for serum 25–OH–D values above 30 ng/mL [29]. Total 25–OH–D levels was calculated as the sum of 25–OH–D_3_ and 25–OH–D_2_ forms.

### 2.5. Statistical Analysis

Qualitative variables were presented as Frequencies (N) and Percentages (%). Quantitative variables were expressed as the arithmetic mean ± Standard Deviation (SD). As a previous step to the execution of a parametric model or not, the hypothesis of normal distribution was accepted using the Kolmogorov-Smirnov test. For the comparative analyses at baseline and follow-up, use was made of the paired Student *t*-test for parametric samples. For the comparative inter-groups analysis, the unpaired Student *t*-test for parametric samples was applied. Categorical variables were compared inter-groups using Chi-square test. Simple linear regression models were conducted to evaluate the significant associations of 25–OH–D levels with biochemical parameters. β (standardized regression coefficient), R^2^, and *p*-Values from simple linear regression analyses were obtained. The SPSS version 26.0 statistical package (IBM SPSS, Armonk, NY, USA) was used throughout, and plots were built using GraphPad Prism 8.0 for Mac (GraphPad Software Inc., San Diego, CA, USA). Statistical significance was considered for *p* < 0.05.

## 3. Results

Table 1 shows the mean changes in the evolution of clinical and biochemical parameters of the study at ICU admission and after 7 days of stay. The mortality rate after 7 days of ICU stay was over one-third of the total study population (38.5%). Only PTH levels were significantly higher in women than in men (*p* < 0.05) at baseline, whereas the rest of the studied parameters showed no differences by sex (*p* ≥ 0.085 in all cases). Similarly, no significant differences were found (*p* ≥ 0.070 in all cases) at baseline after categorizing by median age, except for BMI, being higher at greater ages (*p* < 0.001). Most variables were out of the normal reference ranges at the time of ICU admission, and there was an apparent tendency for the values to normalize after one week of stay. In this line of thinking, and regarding phosphorus–calcium metabolism parameters, the mean changes of osteocalcin, PTH, and Ca levels were significant (*p* > 0.05 in all cases), with an increase in osteocalcin and Ca and a decrease in PTH levels after 7 days of ICU stay.

The evolution of vitamin D and Zn levels during the ICU stay are represented in Table 2. Levels of 25–OH–D were significantly lower than the reference values and remained low after one week, the status being deficient in all cases, at baseline and follow-up. A significant decrease in 25–OH–D_3_ levels (*p* = 0.035) was also found. Regarding Zn status, two-fifths of patients presented hypozincemia at baseline and one-third at follow-up. No significant differences were found in the evolution of plasma and erythrocyte Zn levels throughout the time (all *p* > 0.05). Mean daily support of vitamin D and Zn during the studied period were 4.95 µg/day and 7.75 mg/day, respectively, according to ESPEN recommendations.

Figure 1 shows the associations of vitamin D and Zn levels at baseline and follow-up. Non-significant relationships were found for 25–OH–D, 25–OH–D_3_, and 25–OH–D_2_ with erythrocyte Zn levels and between erythrocyte and plasma Zn at baseline (all *p* ≥ 0.116; panels A, C, E, and G), but the latter became significant at follow-up (all *p* < 0.05; model 0; panels B, D, and H) except for 25–OH–D_2_ (*p* > 0.05, panel F). The significant direct relationship between erythrocyte and plasma Zn levels at follow-up persisted after including age (*p* = 0.022; panel F; Model 1) and sex (*p* = 0.043; panel F; Model 2) as covariates.

Table 3 shows the significant relationships between changes in 25–OH–D, 25–OH–D_3_, and 25–OH–D_2_ levels, and changes in phosphorus–calcium metabolism parameters after one week of ICU stay. The changes in 25–OH–D_3_ levels were found to be directly related to changes in PTH (*p* = 0.019) and P levels (*p* = 0.005).

## 4. Discussion

The main results of the present study showed that critically ill patients presented deficient levels of 25–OH–D at ICU admission, and observed a decrease in 7 days, especially in the case of 25–OH–D_3_. Both 25–OH–D and 25–OH–D_3_ levels were directly associated with erythrocyte Zn concentrations at follow-up, and a relationship also existed between erythrocyte and plasma Zn levels at this time point. No such clear associations were found when considering 25–OH–D_2_. The changes in 25–OH–D_3_ were positively correlated with the changes in PTH and P levels. These findings shed light on the idea that a relationship exists between vitamin D and circulating Zn, mainly due to 25–OH–D_3_ levels.

Patients admitted to the ICU present a critical and complex pathological condition characterized by different clinical and biochemical disturbances, which must be solved for the patient’s recovery [30]. After 7 days of ICU stay, we found significant changes in several biochemical parameters, highlighting an increase in osteocalcin and Ca levels, a decrease in PTH and 25–OH–D_3_ levels, and a reduction in SOFA score. Although certain parameters remained altered, an apparent tendency to stabilize clinical variables was found at follow-up.

Vitamin D status was deficient in all cases at baseline and follow-up. In critically ill patients, vitamin D deficiency has been associated with increased length of stay, morbidity, and mortality, and it has been demonstrated that significant decrease in vitamin D status occurred throughout the patients’ ICU stay [31]. In our study, we observed a significant decrease in 25–OH–D_3_ levels after one week of ICU stay, despite better results observed in severity score at this time point, but the decrease in total 25–OH–D levels was not significant. It should be remembered, however, that without ensuring baseline status and dose adequacy more is not better. A U-shaped curve demonstrated the relationship between 25–OH–D levels (< 30 ng/mL or ≥ 60 ng/mL), and 90-day mortality, and inferred that high-dose vitamin D might not be beneficial to patients [32]. Vitamin D metabolomics, transcriptomics, and epigenetics are needed to investigate whether vitamin D deficiency is an independent contributor to morbidity and mortality of critically ill patients with SIRS and whether “body humors” vitamin D correction leads to outcome benefits [33]. Regarding Zn levels, the obtained mean values of plasma Zn were close to hypozincemia and did not change significantly during the ICU stay [34]. The same tendency was found for erythrocyte Zn levels, which did not change after seven days. Although Zn deficiency is difficult to diagnose, due to the lack of a reliable biomarker to assess Zn status, it is known to be a very common and serious nutritional problem because of the widespread variety of clinical manifestations, especially in developing countries, [35]. In the literature, the advantage of measuring serum Zn levels as one of the routine and standard physiological blood tests has been emphasized [34,36]. Ultimately, having a previous correct status of these micronutrients, through diet, and the use of fortified food or supplements in cases of any risk of deficiency, could be a preventive factor [9,13,37]. No relationships were found at baseline when analyzing the associations between vitamin D and Zn levels. On the contrary, erythrocyte Zn levels were positively correlated to 25–OH–D, 25–OH–D_3_, and plasma Zn levels at follow-up, this last association persisting between circulating Zn levels after age and sex were included as covariates. The influence of vitamin D goes far beyond the regulation of Ca and P balance, playing an important pleiotropic role in maintaining global homeostasis [38]. Vitamin D exerts mitochondrial protective effects through the modulation of different signaling pathways, and downregulates prooxidative and proinflammatory pathways [39]. Several lines of evidence have described the following different interactions between vitamin D and Zn: (i) vitamin D_3_ increases ZnT10 protein expression, (ii) there is a cysteine-rich Zn finger region in the vitamin D receptor, and (iii) vitamin D modulates the expression and activity of Cu/Zn-dependent SOD isoforms and metallothioneins, among others [7,10,19]. Hence, vitamin D could play an important role in Zn-related processes, and it should be investigated more in depth, especially in critically ill patients, where there is a lack of evidence in this line.

Furthermore, the analysis of changes in vitamin D and parameters related to phosphorus–calcium metabolism showed that only the changes in 25–OH–D_3_ levels were directly associated with changes in PTH and P levels. In vitro studies indicate that 25–OH–D_3_ inhibits the synthesis of PTH, and this negative correlation between PTH and 25–OH–D appears to be supported by most of the published studies. However, some authors have suggested that there is a turning point, or even a plateau, in this correlation [40]. Meanwhile, it has been observed that increased P levels may induce hypocalcemia, which increases PTH secretion, and directly stimulate PTH synthesis and parathyroid cellular proliferation [41]. It must be noted that the PTH levels of our cohort were very high at ICU admission and, although they decreased significantly after one week, remained quite high. Likewise, low Ca levels were found at baseline, significantly increasing during the period but not enough to reach normal values. Extracellular ionized Ca concentrations are tightly controlled, and PTH and 1,25–OH_2_–D_3_ are probably the most important factors influencing Ca homeostasis from a clinical point of view [42]. These results are not in line with what would be expected; however, probably these correlations have been affected because of the existence of important disturbances in different studied parameters.

Finally, it should be kept in mind that optimal nutritional status, including healthy levels of vitamin D, is requisite for proper physiological function. Moreover, adequate distribution of metal ions is crucial, and a slight alteration in their levels may induce severe diseases [43]. The interactions between vitamin D and Zn remain incompletely understood. For example, it has been found that Zn supplementation in hemodialysis patients appears to promote bone formation, and is not affected by the status of PTH and vitamin D [44]. Vitamin D_3_ and Zn are known to shift the immune response towards tolerance individually; however, the evidence suggests that their combination may be a promising option to treat dysregulated immune response in various conditions [45]. Therefore, even though more evidence is needed, the correct status of these micronutrients should be ensured as a potentially beneficial approach for the recovery of critically ill patients.

The findings of the present study should be taken with caution as some limitations arise. Firstly, this study recruited fewer patients than desired due to the difficulty of obtaining the sample and the inherent variable clinical situation and severity of the patients. Secondly, follow-up during the 7-day stay in the ICU did not allow us to establish causal relationships. Thirdly, we had no reliable data on exposure to sunlight, dietary factors, or vitamin D supplementation, all of which affect vitamin D status. Fourthly, data on 1,25–OH_2_–D were not available. Fifthly, the overall negative results may be related to the heterogeneity of the subjects and their underlying disease conditions or severity, which may all influence the biochemical profile. Sixthly, we did not find relationships between vitamin D or Zn levels with severity and inflammatory parameters, which could explain the observed differences at baseline and follow-up. As strengths, LC-MS/MS was used to assess the levels of 25–OH–D, which is the gold standard method [46].

## 5. Conclusions

Critically ill patients with SIRS presented deficient vitamin D status. The deficiency decreased after seven days of ICU stay, mainly due to 25–OH–D_3_ levels. The prevalence of hypozincemia decreased at follow-up, but mean plasma Zn levels remained low. A positive association was found between erythrocyte Zn and both 25–OH–D and 25–OH–D_3_ levels at follow-up. Different disturbances in levels of the phosphorus–calcium metabolism parameters were found, signifying a relationship between the changes in 25–OH–D_3_ levels and changes in both PTH and P. The results of the present study show an interaction between vitamin D and Zn after one week of ICU stay, coinciding with the apparent stabilization of critically ill patients. Thus, these findings suggest that the correct status of these micronutrients is a potentially modifiable factor and beneficial approach in the recovery of critically ill patients.

## Figures and Tables

**Figure 1 nutrients-14-03580-f001:**
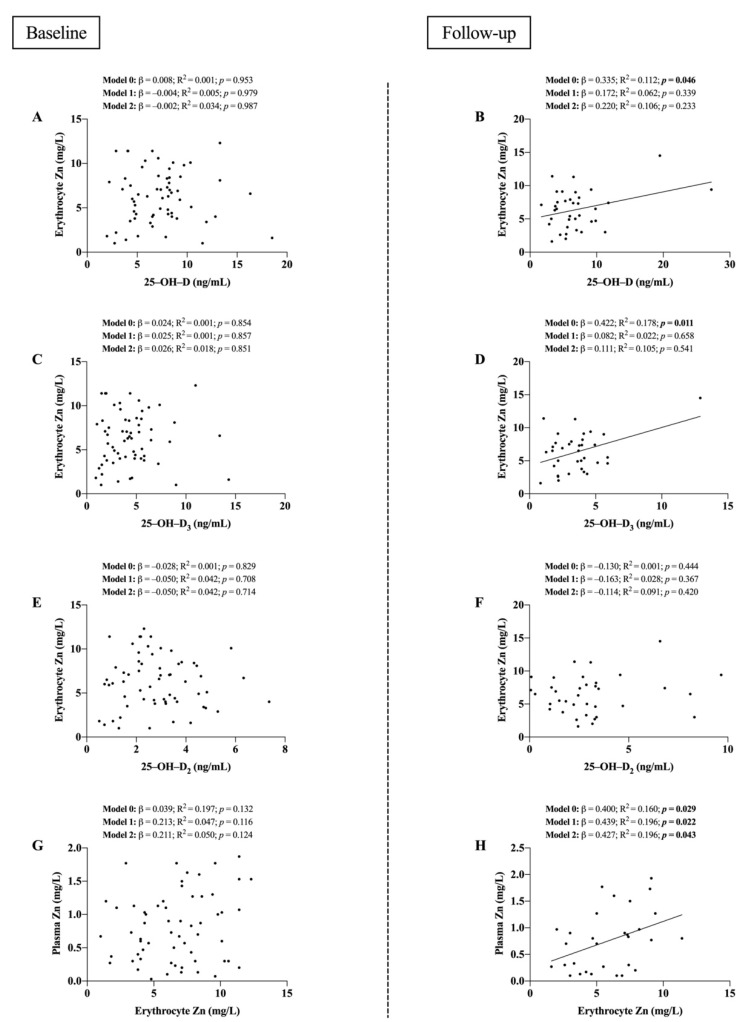
Associations between vitamin D and Zn levels at baseline and follow-up. (**A**) 25–OH–D levels with erythrocyte Zn at baseline. (**B**) 25–OH–D levels with erythrocyte Zn at follow-up. (**C**) 25–OH–D_3_ levels with erythrocyte Zn at baseline. (**D**) 25–OH–D_3_ levels with erythrocyte Zn at follow-up. (**E**) 25–OH–D_2_ levels with erythrocyte Zn at baseline. (**F**) 25–OH–D_2_ levels with erythrocyte Zn at follow-up. (**G**) Erythrocyte Zn with plasma Zn at baseline. (**H**) Erythrocyte Zn with plasma Zn at follow-up. β (standardized regression coefficient), R^2^, and *p* are provided for simple and multiple linear regression analyses. Model 0; unadjusted, Model 1; adjusted for age, Model 2; adjusted for age and sex. Abbreviations: Zn = Zinc; 25–OH–D = 25–Hydroxyvitamin D; 25–OH–D_3_ = 25–Hydroxyvitamin D_3_. Significant *p*-Values are shown in boldface. Statistical significance was considered for *p* < 0.05.

**Table 1 nutrients-14-03580-t001:** Comparative analysis of the studied parameters at baseline and follow-up.

Parameters	Reference	Baseline Mean (SD) *N* = 65	Follow-Up Mean (SD) *N* = 40	Δ Change (%)	*p*-Value Men vs. Women	*p*-Value <62 years vs. >62 years	*p*-Value Baseline vs. Follow-Up
Age (years)	-	60.1 (11.5)	-	-	0.747	-	-
BMI (kg/m^2^)	-	26.7 (4.77)	-	-	0.320	**0.001**	-
Patient 7-day mortality (n/N, %)	-	25/65 (38.5)	-	-	0.474	0.070	-
APACHE-II score	-	17.2 (4.94)	-	-	0.390	0.895	-
SOFA score	-	9.04 (3.39)	5.12 (3.50)	−43.4	0.230	0.264	**<0.001**
Total Proteins (g/dL)	6.60–8.30	5.27 (0.91)	5.53 (0.76)	+4.93	0.138	0.658	0.115
Albumin (g/dL)	3.50–5.20	2.85 (0.58)	2.63 (0.67)	−7.72	0.647	0.649	0.139
Prealbumin (mg/dL)	16.0–42.0	12.5 (4.65)	16.3 (9.80)	+30.3	0.370	0.828	0.143
Ferritin (ng/dL)	20.0–275.0	401.6 (377.4)	543.7 (482.9)	+35.4	0.861	0.493	**0.024**
Transferrin (mg/dL)	200.0–360.0	153.4 (63.1)	146.9 (50.8)	−4.27	0.825	0.868	0.514
Triglycerides (mg/dL)	50.0–200.0	196.9 (144.3)	197.8 (102.1)	+0.46	0.275	0.366	0.972
Total Cholesterol (mg/dL)	140.0–200.0	108.5 (38.2)	134.8 (43.7)	+24.2	0.875	0.521	**0.002**
CRP (mg/L)	0.00–5.00	19.8 (11.8)	10.7 (8.63)	−46.2	0.292	0.164	**0.001**
GOT (U/L)	<37.0	104.0 (164.0)	36.2 (23.2)	−65.2	0.290	0.260	**0.045**
GPT (U/L)	<41.0	54.9 (78.8)	28.5 (26.0)	−48.0	0.801	0.349	0.082
GGT (U/L)	11.0–50.0	56.6 (60.9)	141.3 (98.4)	+149.7	0.932	0.473	**<0.001**
ALP (U/L)	40.0–130.0	101.5 (76.3)	126.1 (68.8)	+24.3	0.085	0.133	0.062
Osteocalcin (ng/mL)	15.0–46.0	2.97 (1.95)	5.53 (3.98)	+86.2	0.385	0.163	**0.002**
PTH (pg/mL)	20.0–70.0	248.5 (151.0)	133.5 (97.9)	−46.3	**0.013**	0.775	**<0.001**
Ca (mg/dL)	8.80–10.6	7.53 (0.86)	8.06 (0.71)	+7.04	0.930	0.812	**0.001**
P (mg/dL)	2.30–4.50	3.91 (1.87)	3.65 (1.47)	−6.65	0.152	0.676	0.452
Mg (mg/dL)	1.60–2.60	2.14 (0.52)	2.31 (0.46)	+7.94	0.564	0.320	0.213

Data are expressed as mean (standard deviation). Abbreviations: ALP = Alkaline Phosphatase; APACHE-II = Acute Physiology and Chronic Health Evaluation II; BMI = Body Mass Index; Ca = Calcium; CRP = C-Reactive Protein; GGT = Gamma-Glutamyl Transferase; GOT = Glutamic Oxaloacetic Transaminase; GPT = Glutamic Pyruvic Transaminase; Mg = Magnesium; P = Phosphorus; PTH = Parathormone; SOFA = Sequential Organ Failure Assessment. Δ Change (%) = percentage change from baseline to follow-up. Qualitative variables were compared inter-groups using Chi-square test. Quantitative variables were compared inter-groups using the unpaired Student *t*-test for parametric samples, and intra-groups using the paired Student *t*-test for parametric samples. Median age (62 years old) was used as cutoff to analyze the differences between lower and higher ages. Significant *p*-Values are shown in boldface. Statistical significance was considered for *p* < 0.05.

**Table 2 nutrients-14-03580-t002:** Comparative analysis of the studied parameters at baseline and follow-up.

Parameters	Reference	Baseline Mean (SD) *N* = 65	Follow-Up Mean (SD) *N* = 40	Δ Change (%)	*p*-Value Baseline vs. Follow-Up
25–OH–D (ng/mL)	30.0–100.0	6.76 (2.96)	6.51 (3.28)	−3.11	0.506
25–OH–D_3_ (ng/mL)	–	3.90 (2.23)	3.59 (2.10)	−7.18	**0.035**
25–OH–D_2_ (ng/mL)	–	2.87 (1.60)	2.93 (2.05)	+2.09	0.875
Plasma Zn (mg/L)	0.65–1.11	0.71 (0.51)	0.74 (0.56)	+4.23	0.748
Erythrocyte Zn (mg/L)	–	6.22 (2.81)	6.42 (2.90)	+3.12	0.708
		**n/N (%)**	**n/N (%)**		
25–OH–D deficiency	<20 ng/mL	65/65 (100)	40/40 (100)	0	-
Hypozincemia (plasma Zn)	<0.7 mg/L	28/65 (43)	13/40 (32.5)	−24.4	**0.001**

Data are expressed as mean (Standard Deviation). Abbreviations: Zn = Zinc; 25–OH–D = 25-Hydroxyvitamin D; 25–OH–D_3_ = 25-Hydroxyvitamin D_3_; 25–OH–D_2_ = 25-Hydroxyvitamin D_2_. Δ Change (%) = percentage change from baseline to follow-up. n = number of participants who presented deficient biochemical values. For the comparative analysis of quantitative variables, the paired Student *t*-test for parametric samples was used. Qualitative variables were compared inter-groups using Chi-square test. Significant *p*-Values are shown in boldface. Statistical significance was considered for *p* < 0.05.

**Table 3 nutrients-14-03580-t003:** Associations between changes in vitamin D and changes in phosphorus–calcium metabolism parameters after 7 days of ICU stay.

Characteristics	Δ 25–OH–D			Δ 25–OH–D_3_			Δ 25–OH–D_2_		
	ß	R^2^	*p*-Value	ß	R^2^	*p*-Value	ß	R^2^	*p*-Value
**Δ Osteocalcin (ng/mL)**	0.091	0.008	0.672	−0.260	0.067	0.221	0.202	0.041	0.343
**Δ PTH (pg/mL)**	0.211	0.044	0.334	0.484	0.234	**0.019**	−0.015	0.001	0.945
**Δ Ca (mg/dL)**	−0.256	0.065	0.218	0.044	0.002	0.833	−0.256	0.065	0.218
**Δ P (mg/dL)**	0.329	0.108	0.109	0.539	0.291	**0.005**	0.115	0.013	0.585
**Δ Mg (mg/dL)**	−0.388	0.150	0.061	0.069	0.005	0.748	−0.402	0.161	0.052

ß = standardized regression coefficient. R^2^ and *p* are from simple regression analysis. Abbreviations: Mg = Magnesium; P = Phosphorus; PTH = Parathormone; 25–OH–D = 25–Hydroxyvitamin D; 25–OH–D_3_ = 25–Hydroxyvitamin D_3_; 25–OH–D_2_ = 25–Hydroxyvitamin D_2_. Bold numbers indicate a statistically significant association. Significance was set at *p*-Value < 0.05.

## Data Availability

The datasets generated and analyzed during the current study are not publicly available due the database is very extensive and includes data from other studies complementary to this but are available from the corresponding authors on reasonable request.

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
