# Peer review of "Vitamin D Status in Critically Ill Patients with SIRS and Its Relationship with Circulating Zn and Related Parameters during ICU Stay"

_nutrients, 2022, doi:10.3390/nu14173580_

Round 1

Reviewer 1 Report

The article needs improvement.

1. Add subchapter title Analytical determination of vitamin Zn. The subsection describing the methodologies for determining Zn.

2. Line 137 correct unit to ° C.

3. Verse 141 quotation of literature should be at the end of the sentence.

4. Verse 247quotation of literature should be at the end of the sentence.

5. Verse 260 quotation of literature should be at the end of the sentence.

6. Verse 141 quotation of literature should be at the end of the sentence.

7. Verse 276 quotation of literature should be at the end of the sentence.

8. Verse 277 quotation of literature should be at the end of the sentence.

9. Verse 278  quotation of literature should be at the end of the sentence.

10. Verse 314 correct the entry 1,25-OH2-D.

Reviewer 2 Report

Critically ill patients  generate systemic inflammatory response syndrome that vitamin D or Zn could be crucial in patients' recovery. The paper reveals that  an interaction between vitamin D and Zn, being a correct status of these micronutrients a potentially modifiable factor and beneficial approach in critical ill patients' recovery.  This paper has great theoretical significance and high practical value for the nutritional structure improvement in critically ill patients. The analysis process is comprehensive and the article is organized, smooth language and so on. Major revision can be published in Nutrients. However, there are some major issues need to be improved:

1. Abstract: The abstract should be modified to enhance the readability;

2. Introduction: Recent references on foods with high vitamin D and Zn content to increase readership;

3. Materials and Methods: Nutritional profile, whether it can be clearly specific?

4. Results: Can you supplement the role in the nutritional formula vitamin D and Zn?

5. Discussion: Can we supplement the vitamin D and Zn effect in food?
